# Zebrafish Avatars: Toward Functional Precision Medicine in Low-Grade Serous Ovarian Cancer

**DOI:** 10.3390/cancers16101812

**Published:** 2024-05-09

**Authors:** Charlotte Fieuws, Jan Willem Bek, Bram Parton, Elyne De Neef, Olivier De Wever, Milena Hoorne, Marta F. Estrada, Jo Van Dorpe, Hannelore Denys, Koen Van de Vijver, Kathleen B. M. Claes

**Affiliations:** 1Department of Biomolecular Medicine, Ghent University, 9000 Ghent, Belgium; charlotte.fieuws@ugent.be (C.F.); janwillem.bek@ugent.be (J.W.B.); bram.parton@ugent.be (B.P.);; 2Center for Medical Genetics Ghent, Ghent University Hospital, 9000 Ghent, Belgium; 3Cancer Research Institute Ghent, 9000 Ghent, Belgium; olivier.dewever@ugent.be (O.D.W.); milena.hoorne@uzgent.be (M.H.); jo.vandorpe@uzgent.be (J.V.D.); hannelore.denys@uzgent.be (H.D.); koen.vandevijver@uzgent.be (K.V.d.V.); 4Laboratory of Experimental Cancer Research, Department of Human Structure and Repair, Ghent University, 9000 Ghent, Belgium; 5Department of Medical Oncology, Ghent University Hospital, 9000 Ghent, Belgium; 6Champalimaud Centre of the Unknown, Champalimaud Foundation, 1400-038 Lisbon, Portugal; marta.estrada@research.fchampalimaud.org; 7Department of Pathology, Ghent University Hospital, 9000 Ghent, Belgium

**Keywords:** ovarian cancer, zebrafish xenografts, functional precision medicine, zAvatars, low-grade serous

## Abstract

**Simple Summary:**

Selecting and developing effective therapies for distinct epithelial ovarian cancer subtypes necessitates tumor models that accurately recapitulate the individual characteristics and microenvironmental interactions. Patient-derived tumor models offer a promising approach by preserving the tumor’s integrity, providing a platform for personalized treatment strategies. Zebrafish embryos could be a useful tool for quickly testing potential treatments in parallel. In this research article, we aimed to evaluate the model using a real-world case study and compare results with existing in vitro and in vivo models. A patient-derived cell line from a low-grade serous ovarian cancer with a *KRAS* mutation was engrafted in zebrafish embryos. Xenografts were assigned to a five-day treatment with trametinib and/or luminespib, targeting two complementary pathways that have previously shown efficacy in other cancer models of this cell line. The zebrafish model offers several options to analyze the response of the tumor to drug exposure. Here, we evaluated both cell growth and cell death, which showed a significant positive tumor response upon treatment that was amplified further by combining the drugs.

**Abstract:**

Ovarian cancer (OC) is an umbrella term for cancerous malignancies affecting the ovaries, yet treatment options for all subtypes are predominantly derived from high-grade serous ovarian cancer, the largest subgroup. The concept of "functional precision medicine" involves gaining personalized insights on therapy choice, based on direct exposure of patient tissues to drugs. This especially holds promise for rare subtypes like low-grade serous ovarian cancer (LGSOC). This study aims to establish an in vivo model for LGSOC using zebrafish embryos, comparing treatment responses previously observed in mouse PDX models, cell lines and 3D tumor models. To address this goal, a well-characterized patient-derived LGSOC cell line with the *KRAS* mutation c.35 G>T (p.(Gly12Val)) was used. Fluorescently labeled tumor cells were injected into the perivitelline space of 2 days’ post-fertilization zebrafish embryos. At 1 day post-injection, xenografts were assessed for tumor size, followed by random allocation into treatment groups with trametinib, luminespib and trametinib + luminespib. Subsequently, xenografts were euthanized and analyzed for apoptosis and proliferation by confocal microscopy. Tumor cells formed compact tumor masses (*n* = 84) in vivo, with clear Ki67 staining, indicating proliferation. Zebrafish xenografts exhibited sensitivity to trametinib and luminespib, individually or combined, within a two-week period, establishing them as a rapid and complementary tool to existing in vitro and in vivo models for evaluating targeted therapies in LGSOC.

## 1. Introduction

Among gynecologic malignancies, ovarian cancer is the leading cause of death due to the late presentation of vague abdominal symptoms and diagnoses at advanced stages. Yet, ovarian cancer is an umbrella term for distinct cancerous malignancies affecting the ovaries, each characterized by its own histology, growth patterns and genetic features. Most research efforts have focused on high-grade serous ovarian cancer (HGSOC), which constitutes the largest subgroup (70–75%) and is associated with poor prognosis. Due to a lack of separate clinical trials, treatment options for all ovarian cancer subtypes including LGSOC have generally been derived from HGSOC trials, despite them being completely different diseases [1]. 

Yet, in the realm of precision oncology, the quest is not merely to classify cancers by their tissue of origin but to dissect them at the molecular level to optimize treatment strategies. Moving away from “one-size-fits-all” therapy regimens raised high hopes for drastic improvements in patient outcome [2]. However, many tumors with predicted genetic vulnerabilities fail to respond to the matched targeted therapy. It is becoming clear that in many cancer phenotypes, in addition to the mutation-driven mechanisms, response to therapy is determined by non-genetic mechanisms which differ from patient to patient [3]. 

These insights led to a new, emerging concept of “functional precision medicine”, an approach based on direct exposure of tumor cells derived from affected individuals to drugs, providing immediately translatable, personalized information to guide therapy [4]. These approaches generate dynamic, functional data that may encompass key vulnerabilities, including those conveyed by altered epigenetic states and/or altered signaling pathways, not necessarily driven by distinct genomic aberrancies [5]. Although conceptually very promising, functional assays are still too premature for clinical practice. Several drug sensitivity screening platforms, both ex vivo and in vivo, have been published and validated. For ex vivo approaches, one can distinguish 2D models (cell lines, primary tumor cells) from 3D models (spheroids, organoids, microfluidics, explants, scaffolds) [6]. These models have the common goal of investigating the response of a cancer to a (combination of) treatment(s). When selecting a preclinical model, several parameters are to be considered, such as experimental duration, long-term effects, the intricacies of the tumor and the inclusion of the Tumor Microenvironment (TME), something that is difficult to recapitulate in most in vitro models [7,8].

Mouse PDX models represent the gold standard for assessing tumor growth and testing of anticancer compounds for ovarian cancer patient-derived in vivo models. However, limitations include ethical controversy, low-throughput drug optimization and animal expenses. The most crucial factor hampering clinical implementation is the long experimental duration to expand and maintain mouse PDX lines [9,10]. Therefore, novel and complementary drug discovery or repurposing platforms are essential. Here, we introduce an emerging in vivo cancer model, zebrafish (Danio rerio). The strengths of zebrafish as model organisms are their high fecundity and rapid external development, as well as easy, low-cost maintenance. Next, zebrafish larvae lack a mature adaptive immune system and are very suited for the generation of patient-derived xenograft models: so-called zebrafish avatars (zAvatars) [7,11,12]. Their optical transparency enables researchers to directly visualize tumor progression, metastasis and microenvironmental interactions such as attraction of innate immune cells and angiogenesis [13,14]. Imaging techniques such as confocal microscopy offer information at single-cell resolution, and semi-automated systems (Operatta CLS^®^) with specialized multiwell plates (ZF plate^®^, Hashimoto) can be used for real time and high-resolution imaging [15,16]. 

In contrast with other cancer types, only a few research groups have explored the utility of zAvatars for functional personalized medicine in ovarian cancer, and thus far, none have focused on low-grade serous ovarian cancer (LGSOC) [17,18]. LGSOC is relatively resistant to standard chemotherapy, likely due to its low proliferative activity in comparison with HGSOC. However, LGSOC often shows an activated mitogen-activated protein kinase pathway (MAPK) by oncogenic *KRAS*, *NRAS* or *BRAF* mutations, which are effective biomarkers that can be exploited therapeutically [19]. 

The aim of this study is to establish an in vivo drug screening model for LGSOC using zebrafish larvae. The primary objective is to compare the response to targeted therapies in this zebrafish model to the response observed in other previously published models including mouse PDX, cell lines and 3D tumor models [20,21]. To address this goal, we used a well-characterized cell line, established from an early-stage transplantable peritoneal metastasis-mouse PDX model (PM-PDX) of an LGSOC with the *KRAS* mutation c.35 G>T (p.(Gly12Val)). Although this cell line is known to be sensitive to the MEK inhibitor trametinib, clinical trials have demonstrated the occurrence of MEKi resistance in LGSOC patients [22,23]. Therefore, to enhance therapy durability, the potential of drug combinations has previously been explored with this cell line. Specifically, Heat Shock Protein (HSP)90 inhibitors are promising agents to combine with MEKi, because these drugs target a complementary pathway, hereby markedly reducing AKT and mTOR phosphorylation. The combination of MEKi (trametinib) and HSP90i (luminespib) treatment showed delayed tumor formation in scaffolds, an in vitro 3D tumor model and increased survival in an orthotopic LGSOC xenograft mouse model [20]. 

This study seeks to demonstrate the potential of zebrafish xenografts as complementary tools to existing in vitro and in vivo tumor models. On top of that, we provide an exclusive insight into the technical challenges and pitfalls of using this xenograft model in cancer research. 

## 2. Materials and Methods

### 2.1. Animal Care and Handling

Adult zebrafish were housed in a semi-closed recirculating system (ZebTek, Techniplast, Milan, Italy) at a temperature between 27 °C and 28 °C, conductivity ~500 µS, pH 7.5 and 14 h light/10 h dark cycle (Ghent zebrafish facility: https://www.zebrafishfacilityghent.org/, accessed on 18 March 2024). Maintenance and handling of zebrafish were executed as recommended by EU directive (2010/63/EU) for animals. Transparent zebrafish line Casper (roy^−/−^; nacre ^−/−^) was used for establishment of zebrafish xenografts [14]. Egg spawning and collection of embryos were performed according to Westerfield [24]. Zebrafish embryos were kept in a separate incubator in E3 medium at 28 °C until injection. All studies and procedures were approved by the local animal ethical committee (Ghent University Hospital, Ghent, Belgium), permit no. ECD 19-87. 

### 2.2. Cell Culture

The cell line applied in this study was derived from an orthotopic PDX mouse model, obtained from an early-stage peritoneal metastasis in a patient with low-grade serous ovarian cancer (PM-LGSOC-01). The establishment of this cell line has been described extensively by De Thaye et al [21]. The cells were cultured in Eagle’s Minimum Essential Medium (EMEM) (10-009-CV, Corning, NY, USA), supplemented with 10% heat-inactivated fetal bovine serum (FBS), 100 IU/ml penicillin and 100 mg/ml streptomycin (15070063, ThermoFisher, Waltham, MA, USA). Cells were expanded and maintained as a monolayer at 37 °C in an atmosphere 5% CO_2_ (LGSOCs) in air and passaged at 80% confluence. Cells were tested for mycoplasma contamination using Mycoalert Mycoplasma Detection Kit (LT07-318, Lonza, Verviers, Belgium).

### 2.3. Establishment of Zebrafish Avatars

Zebrafish xenografts were generated according to a protocol previously described and visualized by Fior et al [25]. LGSOC cells were cultured in T75 flasks until 70% confluence was reached. The cells were then fluorescently stained with lipophilic membrane dye, Vybrant CM-DiI (ThermoFisher Scientific, Waltham, MA, USA) at a concentration of 4 µL/mL in PBS. Cells were detached from the surface of the flask using 50 mM EDTA in PBS and a cell scraper. Cells were centrifuged (300 RCF, 5 min) and resuspended in PBS to a final concentration of 0.25 × 10^6^ cells/µL and kept on ice. Before injection, 2 dpf zebrafish embryos were anesthetized with Tricaine 1× (Sigma-Aldrich, Burlington, MA, USA) for 5 minutes. The labeled cells were subsequently injected into the perivitelline space (PVS) of zebrafish embryos using a FemtoJet^®^ 4i microinjector (Eppendorf, Hamburg, Germany). Tumor cells were injected until a tumor, with a size comparable to the eye of the zebrafish embryo, was reached. Zebrafish xenografts were kept sedated for 10 min after injection before they were transferred to E3 medium and kept at 34 °C until the end of the experiments. Twenty-four hours post-injection (hpi), zebrafish xenografts were screened for tumor size and correct tumor location (in PVS) with Nikon SMZ18 stereoscope (RFP; 3.2× magnification). Xenografts suffering from edema, xenografts with tumor cells exclusively in the yolk sac and poorly injected zebrafish embryos were discarded and euthanized. Correctly injected zebrafish xenografts were randomly distributed into 4 treatment groups: (1) 100 nM trametinib (C988930 Bioconnect), (2) 1 µM luminespib (ORB154741, Bioconnect), (3) Combination 100 nM trametinib and 1 µM luminespib, (4) Negative control (0.1% DMSO in E3 medium). Treatment dosages were derived from in vitro scaffold experiments conducted by De Vlieghere et al [20]. In vivo concentrations in zebrafish were 100 times’ greater than those employed in vitro. All treatments were assembled by addition of the compound to E3 medium as swimming solution and were refreshed daily for 4 consecutive days. Xenografts were kept separately in a 48-well plate. At 5 days post-injection (dpi), zebrafish xenografts were euthanized with tricaine 25× for postmortem analysis.

### 2.4. Immunohistochemical Staining

For immunohistochemical analysis, the euthanized xenografts were fixed with 4% paraformaldehyde (PFA) overnight at 4 °C and subsequently embedded in paraffin. Zebrafish xenografts, fixed and paraffin-embedded for IHC, were sectioned at 5 µm and stained with hematoxylin and eosin (H&E) to compare the morphology and growth patterns with the original primary tumor, or stained for Ki67 to estimate the proliferative capacity of the implants. Images were obtained with a Zeiss Axio Observer.Z1 inverted microscope using 10× and 40× objectives, and processed by Zen pro 2012 software.

### 2.5. Whole Mount Immunofluorescent Staining

Preceding whole mount staining, the embryos were fixed with 4% PFA + 0.1% Triton X-100 (Sigma-Aldrich, Burlington, MA, USA) for 4 hours at room temperature (RT), then transferred to 100% methanol and stored at −20 °C until start of the whole mount staining. The xenografts were first rehydrated by a series of decreasing methanol concentrations (75% > 50% > 25% > 0%) diluted in PBS/triton 0.1%. The final step of PBS/triton 0.1% followed by a short incubation in ice cold acetone 100% allowed permeabilization. Xenografts were blocked for 1 h at RT with blocking buffer (0.01g/mL BSA, 1% DMSO, 1% triton X-100, and 0.0225% normal goat serum diluted in 1× PBS). Next, xenografts were incubated overnight at 4 °C with a primary antibody: rabbit anti-cleaved caspase-3 (9661S, Cell Signaling, Danvers, MA, USA) or rabbit anti-Ki67 (ab15580, Abcam, Cambridge, UK) diluted at 1:100 in blocking buffer. Next day, xenografts were washed multiple times with PBS/Triton 0.1% and incubated overnight at 4 °C with a secondary antibody: Alexa Fluor^®^ 488 goat anti-rabbit IgG (Life Technologies, Foster City, CA, USA) diluted at 1:400 in blocking buffer, and 10 µg/mL DAPI (Sigma-Aldrich, Burlington, MA, USA) as counterstaining for nuclei. After whole mount staining, xenografts were washed thoroughly and fixed with 4% PFA. They were mounted with Mowiol (Sigma-Aldrich, Burlington, MA, USA) between two coverslips for confocal imaging. Images were obtained using a Zeiss spinning disk system with 5 µM Z-stack interval function and 25× water objective. Images were analyzed using ZEN blue version 3.6 and FIJI/ImageJ version 1.53c software.

### 2.6. Cleaved Caspase-3 Quantification

The total number of cleaved caspase-3 positive cells in the tumors were counted manually in every Z-stack using Cell Counter plug in for FIJI/ImageJ version 1.53c software and adjusted ratios were calculated by dividing positive cell count by tumor volume. Tumor volume of each xenograft was calculated as follows: (1)Define the ROI of all Z-stacks and measure the area in FIJI/ImageJ,(2)(Calculate the volume of every Z-stack: area ROI × Z-stack size,(3)Tumor volume = sum of Z-stacks volume.

### 2.7. Ki-67 Quantification

Due to the density of the tumor cells, it was impossible to quantify Ki67 expression at single-cell level. Therefore, an alternative approach was used based on binary masking in FIJI/ImageJ version 1.53c:(1)Generation of Maximum Intensity Projections (MIPs) from relevant Z-stacks were generated using ImageJ,(2)Transformation of 8-bit files to RGB file,(3)Binary masking of Ki67 staining using fixed threshold,(4)Select Region of Interest (ROI) of tumor,(5)Percentage of stained area in ROI.

### 2.8. Statistical Analysis

Statistical analysis was performed using GraphPad Prism (version 10.2.0, GraphPad Software). Multiple comparisons between treatment groups were determined by One-way ANOVA; Tukey procedure was used to adjust *p*-values. *p*-values of 0.05 were considered as statistically significantly different and output was represented as: not-significant (ns) *p* > 0.05, * *p* ≤ 0.05, ** *p* ≤ 0.01, *** *p* ≤ 0.001 and **** *p* ≤ 0.0001.

## 3. Results

### 3.1. Establishment of Zebrafish Avatars from LGSOC Cells

zAvatars were created by injection of LGSOC single-cell suspensions from early passage cultures (PM-LGSOC-01) into the perivitelline space (PVS) of 2 days’ post-fertilization (dpf) zebrafish embryos (Figure 1a). Xenografts were screened daily by a Nikon SMZ18 stereoscope (RFP; 3.2× magnification) to evaluate tumor progression based on red fluorescent staining of tumor cells. At 5 days’ post-injection (dpi), no significant difference in tumor size could be measured compared to 1 dpi (Figure 1b,b’). Hematoxylin and eosin (H&E)-stained xenografted tumors revealed a micropapillary to focal solid growth pattern typical for LGSOC and comparable to the original patient tissue and mouse PDX (Figure 1c,c’) [21]. Histochemical and whole mount staining for Ki67 showed the proliferative capacity of the LGSOC cells in the zebrafish larvae (Figure 1d,e).

### 3.2. MEKi and HSP90i Efficacy in LGSOC Zebrafish Avatars

In total, 84 xenografts were available for compound screening. The embryos were randomly distributed into four treatment conditions (control, trametinib, luminespib, trametinib + luminespib). Next, drug-response was assessed by measuring both proliferation and apoptosis. Hereto, immunohistochemical and whole mount immune fluorescent staining for Ki67 and cleaved-caspase-3 markers, respectively, were applied postmortem. To assess the immediate cytotoxic effects of trametinib and/or luminespib, zebrafish xenografts were treated for 96 h, euthanized for immune fluorescent staining and analyzed using a confocal microscope. 

Whole mount staining for Ki67 (Figure 2) indicated decreased proliferation upon treatment with trametinib (** *p* = 0,0011) and luminespib (* *p* = 0.0395); combining trametinib + luminespib resulted in even lower Ki67 expression (**** *p* < 0.0001) (Figure 2i).

Cells in apoptosis were quantified by measuring the expression of cleaved caspase-3. For single compound conditions, no significant differences in cleaved caspase-3 were observed compared to the control. Remarkably, the combined trametinib and luminespib treatment induced higher levels of cleaved caspase-3 compared to the control group (*** *p* = 0.0009), trametinib (* *p* = 0.0186) and luminespib (ns *p* = 0.0511) alone (Figure 3a–g).

## 4. Discussion

Ovarian cancer is a heterogenous disease with phenotypic and genetic variations that can impact treatment responses among patients [26]. Selecting an effective therapy for each patient necessitates models that accurately recapitulate the tumor’s characteristics and microenvironmental interactions. Patient-derived tumor models offer a promising approach by preserving the tumor integrity, providing a platform for personalized treatment strategies. Major efforts in preclinical models, such as PDX in mice or organoids, are focused on HGSOC, which constitutes the largest subgroup of ovarian cancers. 

In this study, we focused on an LGSOC with a *KRAS* mutation from which several well-characterized patient-derived tumor models are established, generated from peritoneal metastases, including a patient-derived cell line, mouse PDX and long-term 3D scaffold [20,21]. All models predicted sensitivity to trametinib and luminespib, either as solo or combination treatment. The limitation of these models is the long experimental duration (2–3 months), which prevents its use in clinical decision making, as the median time between diagnosis and treatment should be less than a month [27]. Therefore, the goal of this research was to investigate whether patient-derived zebrafish avatars allow us to stratify patients for treatments in a more clinically relevant timeframe, using a well-described patient-derived LGSOC cell line. 

During the past decade, zebrafish avatars entered the world of functional precision medicine from proof-of-concept studies involving a diverse set of cancer cell lines, evolving to experiments with zebrafish PDX models and, currently, the first co-clinical trials [17]. To date, nearly a thousand research papers have been published on the utility of zebrafish avatars in predicting tumor responses. Their popularity stems from numerous advantages including easy and low-cost maintenance, high sample-size experiments and a diverse range of read-out possibilities such as whole mount fluorescent immunostaining. 

Various protocols are described for both the generation of zebrafish xenografts and downstream analysis. For instance, possible injection sites are determined by the skill of the researcher and should be relevant to the type of cancer, whereby orthotopic locations (eyes, brain, duct of Cuvier) are preferred [28]. If an orthotopic location is not possible, it has been demonstrated that the optimal location in zebrafish embryos is the perivitelline space (PVS), which can accommodate a large number of cells, allows for vascularization and facilitates easy imaging. However, injection in this small space between the periderm and yolk syncytial layer requires considerable expertise and practice. Yet the yolk sac is often chosen as a more practical alternative, despite being suboptimal in engraftment efficiency and less biologically relevant (no vascularization, low macrophage infiltration) [29]. Based on this evidence, we opted for PVS injection and demonstrated successful engraftment, persistence and proliferation of the patient-derived cell line during the time-window of the experiment.

Most published work on zebrafish xenografting is based on analyzing tumor size, e.g., measuring the area or volume of the cells via a fluorescent signal [15,17]. However, we find that estimating tumor response is insufficient and inaccurate if only relying on changes in dye-based cell labeling. First, the most commonly used fluorescent stain, Vybrant CM-DiI, remains visible in dead tumor cells, making it impossible to distinguish healthy and dividing cells from cells undergoing stress or apoptosis (Appendix A). It was recently shown that other staining protocols such as the cytoplasmic stain CellTraceTM Violet, might be more representative of actual live tumor cells [15]. This, however, does not solve a second problem when analyzing tumor volume and/or size. Due to absorption of the yolk sac between 3 dpf and 7 dpf, the engrafted tumor mass will slowly change shape and location over time, hindering the objective measurement and comparison of tumor size between 1 dpi and 5 dpi. Only single-cell high-resolution confocal microscopy is able to circumvent this problem [16]. However, daily imaging of large sample sizes in a specific orientation is time-consuming. While specific multi-well plates (ZF-plate^®^, Hashimoto) are available to sort and position zebrafish with the possibility of fluorescent imaging, they do not address the other drawbacks of live imaging, including anesthesia induced mortality, lower incubation temperature and no exposure to treatment during imaging procedures [15].

Therefore, we opted to evaluate tumor response, via a postmortem read-out to identify pre-apoptotic features. First, we used a paraffin-embedding tissue approach allowing for multiple histochemical stainings. H&E staining revealed growth patterns similar to those observed in the primary tumor and mouse PDX [21]. Immune histochemical (IHC) staining for Ki67 demonstrated the proliferative capacity of the tumor cells within their host zebrafish. Since section-based histology is time-consuming, whole mount staining was employed, offering a more streamlined method that allows for larger sample sizes. Confocal microscopy was applied, providing optical sectioning and high resolution. Quantification can be performed by dividing marker positive cells by the total tumor cell count, as previously described [30]. However, despite the expected single-cell resolution, tumor cells were too dense, and nuclei could not be distinguished from each other in a three-dimensional setting. This highlights the problem that even when performing injections in biologically relevant anatomic locations, image quality can be insufficient due to limitations of the available imaging modalities. Therefore, we measured the total tumor volume instead of the total cell count.

Cleaved caspase-3 is a key player in proteolysis during apoptosis. It is hypothesized that expression of caspase should be higher in tumors treated with effective drugs [31]. In the zebrafish xenografts, tumor cells positive for cleaved caspase-3 could be counted at single-cell level throughout the entire tumor, without identifying hotspot regions. The combination of trametinib with luminespib induced high levels of cleaved caspase-3 compared to solo treatment conditions with these compounds, while in the control group, minimal cell death occurred. The slight elevation of caspase-3 expression in solo treatment conditions compared to the control was not statistically significant, and neither were the differences between the two compounds. Since cleaved caspase-3 is an early apoptotic marker, extending the treatment duration could induce more apoptosis resulting in greater differences between the conditions. 

To assess earlier effects in the cell cycle, a second marker, Ki67, was used to quantify the proliferative capacity of tumor cells. Ki67 expression is widely adopted in pathology, particularly in studies assessing its predictive value in therapy outcome in low-grade serous ovarian cancer [32,33]. As Ki67 was highly expressed throughout the tumor in all conditions, single-cell counting as with cleaved caspase-3, was not feasible. Therefore, quantification of Ki67 expression was performed based on binary masking with a fixed threshold, and revealed significant differences between treatment groups. Decreased proliferation upon treatment with trametinib and luminespib was shown; combining trametinib + luminespib resulted in even less proliferation. The standard deviation of the data was higher than for caspase, emphasizing the need for large sample sizes for accurate result interpretation. It is crucial to note that setting the threshold is an arbitrary decision based on a training set of confocal images. It is reasonable to think that the threshold should be adapted for every experiment or staining, but most importantly, it must be fixed within an experiment between different conditions. 

In conclusion, initial observations on tumor size did not reveal effects of drug treatment, potentially leading to the erroneous conclusion that the given treatment is not effective (Appendix A). However, considering proliferation and apoptosis, we demonstrate a significant effect of trametinib and luminespib. Caspase was less abundantly expressed and turned out to be very suitable for accurate quantification, but specifically stains late tumor responses. On the other hand, Ki67 displayed more obvious differences between treatment conditions as early tumor response marker. Therefore, careful application and detailed description of quantification methods is crucial for extrapolating results to independent experiments. The combined quantification of these two markers provides a reliable interpretation of the tumor response to treatment. The observed sensitivity to trametinib and luminespib, either as solo or combination treatment within two weeks, defines zebrafish xenografts as a quick and complementary tool to existing in vitro and in vivo tumor models. To illustrate this, the total duration of the in vivo mouse model experiment with this cell line was 120 days with first effects of the single treatments after 35 days and a stronger effect of trametinib combined with luminespib after almost 70 days. The LGSOC in vitro scaffold model lasted for 50 days with differential response to treatment observed starting from 20 days’ post-treatment [20]. 

The generation of zebrafish xenografts and downstream analysis based on immune staining can easily be obtained in less than two weeks without complex and expensive lab equipment. However, the introduction of more elaborate read-out methods could enhance the translational value of future studies. For example, the use of transgenic fluorescent zebrafish lines enables the visualization and tracking of specific cells and cell interactions in real time, offering insights into various cancer hallmarks. Cancer applications include the visualization of vascularization and angiogenesis by fluorescent endothelial cells (cfr Tg(kdrl: eGFP)), or the visualization of the interaction with the innate immune system by applying zebrafish lines with fluorescent myeloid cells, including macrophages and neutrophils (e.g., Tg(mpeg1:eGFP and Tg(mpx:mCherry) [14]. Furthermore, the metastatic potential of cancer cells can be monitored through fluorescently labeled cells that extravasate to other tissues and travel through the bloodstream. Additionally, molecular insights into tumor responses can potentially be monitored by extracting DNA, RNA or proteins for subsequent analysis in bulk or at single-cell level [13]. 

## 5. Conclusions

In conclusion, while there are numerous read-out opportunities that can be adopted from other cancer models, the unique features intrinsic to the zebrafish model allow for semi-high throughput, less invasive and high-resolution tumor response prediction. Additional co-clinical trials are required, spanning diverse ovarian cancer subtypes, to compare zPDX avatar data with patient treatment responses. In summary, this experimental research serves as a real-world case report, demonstrating that generation of patient-derived zebrafish avatars is feasible and could enable precision medicine in an economic manner and clinically relevant time-frame.

## Figures and Tables

**Figure 1 cancers-16-01812-f001:**
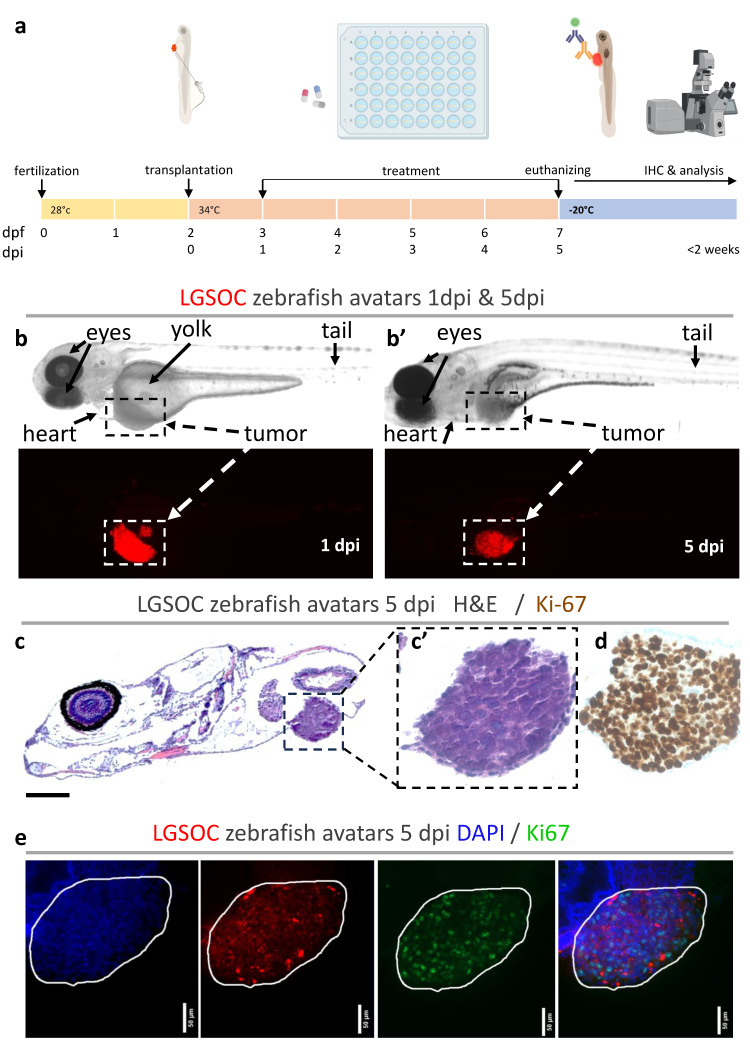
Establishment and analysis of zebrafish xenografts from LGSOC cells. (**a**) Schematic workflow for the generation of zebrafish xenografts from LGSOC cells. Early passage cell line was labeled with Vybrant CM-DiI dye (red) and microinjected in the PVS of 2 dpf larvae. One dpi larvae were screened and randomly distributed into treatment groups. After 96 h of treatment, xenografts were euthanized and divided for postmortem analysis based on immune staining. (**b**,**b’**) Assessment of tumor progression through live-cell imaging in zebrafish xenograft at 1 dpi and 5 dpi. Top row shows brightfield picture; bottom row shows RFP filter with tumor cells expressing red fluorescence. Anatomic structures are indicated in pictures; dashed box indicated tumor area. (**c**,**c’**) Hematoxylin eosin staining and (**d**) immunohistochemistry for Ki67 were performed in histological sections to observe morphology and proliferative capacity of the cancer cells in the zebrafish larvae. Dashed box indicates area of zoom. (**e**) Representative whole mount immunofluorescence staining for Ki67, all images are at 25× magnification (scale bar, 50 µm). Nuclei stained with DAPI in blue, anti-Ki67 in green and fluorescently labeled cancer cells in red. Tumor area is indicated with a white line. All images are anterior to the left, dorsal up.

**Figure 2 cancers-16-01812-f002:**
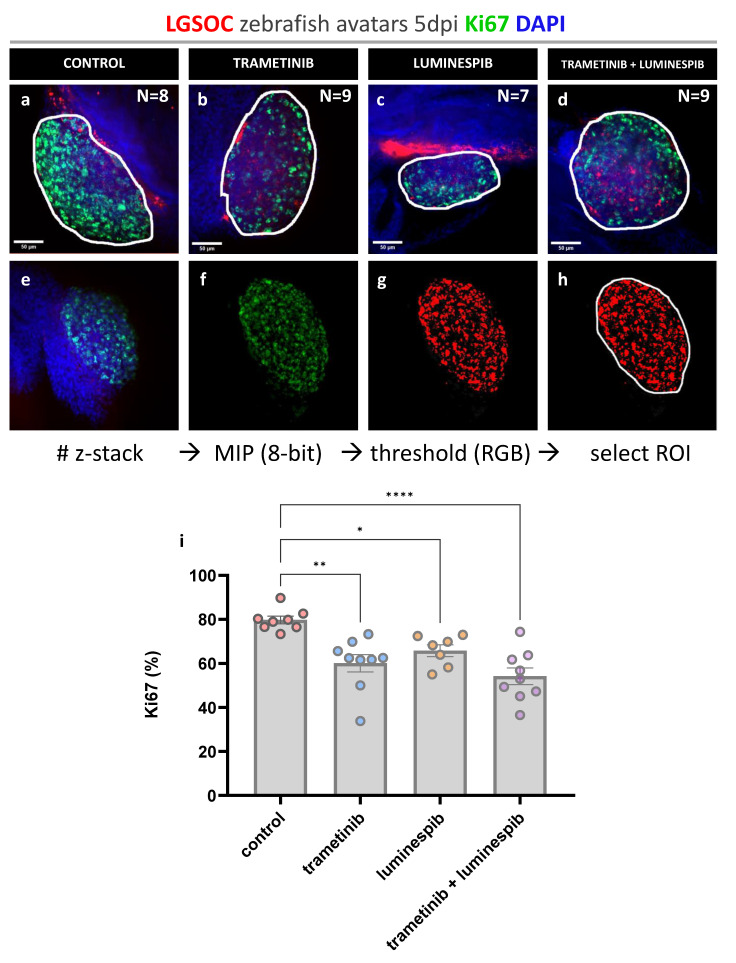
Whole mount immune fluorescence staining for Ki67. Vybrant CM-DiI labeled LGSOC cells (red), nuclei stained with DAPI (blue) and anti Ki-67 (green). (**a**–**d**) The number of xenografts analyzed for Ki67 is indicated in the figures; all images are at 25X magnification (scale bar 50 µM). Tumor area is indicated with a white line. (**e**–**h**) Quantification of Ki67 expression in zebrafish xenografts using ImageJ. Generation of Maximum Intensity Projections (MIPs) from relevant Z-stacks (**e**), transformation of 8-bit files to RGB file (**f**), binary masking of Ki67 staining using fixed threshold (**g**), select ROI of tumor (**h**), percentage of stained area in ROI represents Ki67 expression. (**i**) Percentages of different treatment conditions are presented as AVG ± SEM and each dot represents a xenograft. Multiple comparisons between treatment groups were determined by One-way ANOVA; Tukey procedure was used to adjust p-values. *p*-values of 0.05 were considered as statistically significantly different and output was represented as: not-significant (ns) *p* > 0.05, * *p* ≤ 0.05, ** *p* ≤ 0.01, and **** *p* ≤ 0.0001.

**Figure 3 cancers-16-01812-f003:**
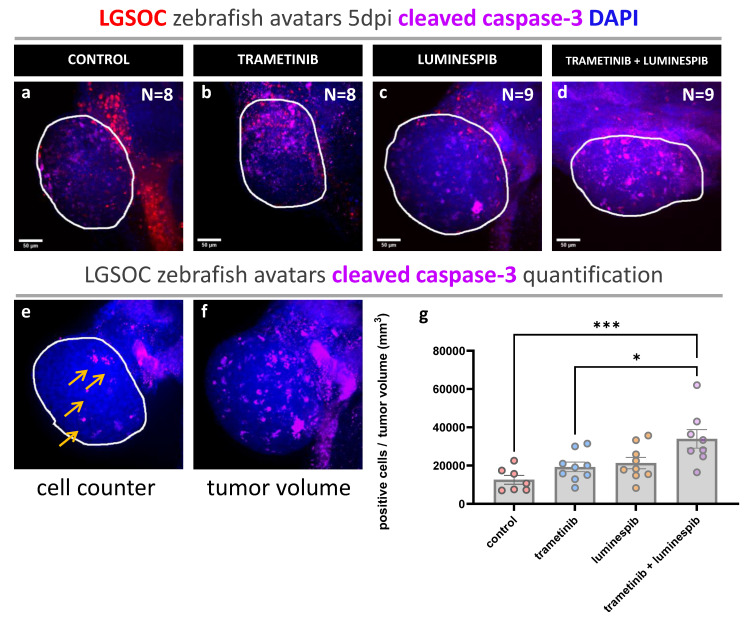
Whole mount immune fluorescence staining cleaved caspase-3. Vybrant CM-DiI labeled LGSOC cells (red), nuclei stained with DAPI (blue) and anti-cleaved caspase-3 (violet). (**a**–**d**) The number of xenografts analyzed for cleaved caspase-3 is indicated in the figures; all images are at 25× magnification (scale bar 50 µM). Tumor area is indicated with a white line. (**e**,**f**) Quantification of cleaved caspase-3 expression in zebrafish xenografts using Cell Counter plugin in ImageJ. Cells positive for cleaved caspase-3 are indicated by yellow arrows. (**g**) Ratios of different treatment conditions are presented as AVG ± SEM and each dot represents one xenograft. Multiple comparisons between treatment groups were determined by One-way ANOVA; Tukey procedure was used to adjust *p*-values. *p*-values of 0.05 were considered as statistically significantly different and output was represented as: * *p* ≤ 0.05, *** *p* ≤ 0.001.

## Data Availability

Confocal microscopy raw imaging data are available upon request.

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
