# Peer review of "Zebrafish Avatars: Toward Functional Precision Medicine in Low-Grade Serous Ovarian Cancer"

_cancers, 2024, doi:10.3390/cancers16101812_

Round 1

Reviewer 1 Report

Comments and Suggestions for Authors

The study by Fieuws et al. investigates the zebrafish xenograft model for low-grade serous ovarian cancer and shows anti-tumor effects of MEKi and HSP90i as single agents and in combination. The authors suggest that a readout based on immunostaining for Ki67 and cleaved Casp-3 is superior to area or volume measurements, but a direct comparison is missing. The manuscript is well written and the experiments are generally well performed. There are several points for further improvement of this study.

Major

-          The summary/introduction suggests that this study will compare drug treatment results from zebrafish with in vitro and other in vivo models. This would be very valuable to the field. Unfortunately, no actual comparison is presented in the results part, which makes this study less exciting.

-          In Figure 2 and more so in Figure 3, several DiI-labeled tumor cells appear to be outside the labeled tumor region. How is the tumor region determined?

-          There seems to be caspase-3 pos. cells outside the circled tumor region in Fig.3b-d as well. Are the authors using a human specific antibody?

-          The authors claim in the discussion that counting Ki67 or Caspase positive cells is superior to a tumor volume or area-based readout, but they do not show any data. They should perform this comparison and show the data to support this claim.

-          The authors mention in the discussion that there is quite same variability concerning treatment response and a large sample size is required for accurate results. However, the experiments seem to be done as just one technical replicate? If this is the case, the authors should perform a second experiment to confirm their results.

-          Line 266, the authors claim a suggested synergistic effect. It would strengthen their claim to show this synergy formally, e.g. by using the Bliss independence model in vitro.

-          In their conclusions statement, the authors claim they demonstrated that zebrafish avatars enable the stratification of patients for treatments, but they only used one cell line in this study. To make such a claim, several patient-derived zebrafish xenografts for LGSOC need to be established.

Minor

-          The authors state that the LGSOC cells are highly proliferative in the fish and they see hardly any apoptotic cells without treatment. How do they explain that there is no increase in tumor size (see Fig. 1b)? It would be good to determine the doubling time of the cells in vitro at 34°C.

-          It would be great to add an explanation, why the authors opted for the drug concentrations applied for the xenograft treatment. 100x in vitro concentrations are not so common.

-          In the methods, the authors mention 10x, 40x, 25x oculars, however, I believe they actually refer to objectives and not oculars

-          Line 265, it should say Figure 3 instead of Figure 1.

-          The figure legend of the second figure should be labeled with figure 2.

-          In Fig.3 h the authors show a calculated number of casp3 pos. cells per tumor volume.

It would be great to explain how the tumor volume was calculated in more detail. It is not so clear from the methods (ROI x Z stack size). How does that account for more complex shaped tumors where the area per z plane changes a lot throughout the stack?

-          What is the throughput of this immunostaining based quantification method?

-          It would be interesting to compare the number of cells at 1dpi with 5dpi. Also a comparison of Ki67 pos. cells between these time points would be of interest.

Reviewer 2 Report

Comments and Suggestions for Authors The article shows the experimental possibilities of response to therapy of low-grade serous tumors using the zebrafish avatar model. The study is very well designed, especially in view of the fact that low-grade serous carcinoma responds poorly to therapy. The methods are described in detail and concisely and results are well-illustrated graphically. The authors have listed the latest literature on the subject. Such research is encouraging and promising for the development and improvement of therapy for these rare but therapeutically problematic tumors. Minor suggestion:
A low-grade serous carcinoma is not simply a low-grade form of a high-grade serous tumor. It is a completely different disease that differs from high-grade serous carcinoma in terms of pathogenesis, molecular, genetic and clinical features. This should be emphasized more strongly in the introduction or the discussion. Comments on the Quality of English Language The spelling is inconsistent (there is a mixture of US and UK English). In my opinion, academic articles should maintain consistent spelling. In general, it is probably acceptable to use either US or UK English, but not a mixture.

Round 2

Reviewer 1 Report

Comments and Suggestions for Authors

The authors addressed my points of criticism.